# New Horizon in Selective Tocols Extraction from Deodorizer Distillates Under Mild Conditions by Using Deep Eutectic Solvents

**DOI:** 10.3390/molecules30061217

**Published:** 2025-03-08

**Authors:** Dian Maria Ulfa, Asep Bayu, Siti Irma Rahmawati, Peni Ahmadi, Masteria Yunovilsa Putra, Surachai Karnjanakom, Guoqing Guan, Abdul Mun’im

**Affiliations:** 1Faculty of Pharmacy, Cluster of Health Sciences Building, Universitas Indonesia, Depok 16424, West Java, Indonesia; dian.maria@poltekkesjkt2.ac.id; 2National Metabolomics Collaborative Research Center, Faculty of Pharmacy, Cluster of Health Sciences Building, Universitas Indonesia, Depok 16424, West Java, Indonesia; siti058@brin.go.id (S.I.R.); peni.ahmadi@brin.go.id (P.A.); masteria.yunovilsa.putra@brin.go.id (M.Y.P.); 3Health Polytechnic Jakarta II, South Jakarta 12540, Jakarta, Indonesia; 4Research Center for Vaccine and Drugs, National Research and Innovation Agency (BRIN), Jalan Raya Jakarta-Bogor KM 46, Cibinong, Bogor 16911, West Java, Indonesia; 5Department of Chemistry, Faculty of Science, Rangsit University, Pathumthani 12000, Thailand; surachai.ka@rsu.ac.th; 6Energy Conversion Engineering Laboratory, Institute of Regional Innovation, Hirosaki University, 3-Bunkyocho, Hirosaki 036-8561, Japan; guan@hirosaki-u.ac.jp

**Keywords:** tocotrienol, tocopherol, extraction, deodorizer distillates, deep eutectic solvents

## Abstract

Tocols are commonly known as vitamin E, which comprise tocopherols and tocotrienols. Although vegetable oils are natural sources of tocols, deodorizer distillates (DDs) are attractive feedstock due to their potential abundance from oil refining processes and economic price. Deep eutectic solvents (DESs) are a family of neoteric solvents that show promising performance for tocols extraction. Besides their characters occupying the green chemistry concept, this review presents the current research on the potential performances of DESs in extracting tocols selectively and efficiently from DDs. The application of DESs in tocols extraction is presented considering three different ways: mono-phasic, in situ DESs formation, and bi-phasic systems. The basic principles of intermolecular interactions (H-bond, van der Walls bond, and misfit interaction) between DESs or their components with tocols are discussed to understand the mechanism by which DESs selectively extract tocols from the mixture. This is mainly observed to be a function of the intrinsic properties of DESs and/or tocols, which could be beneficial for tuning the appropriate DESs for extracting tocols selectively and effectively under mild operation conditions. This review is expected to provide insight in the potential application of DESs in the extracting of natural compounds with a phenolic structure and also briefly discusses the toxicity of DESs.

## 1. Introduction

Tocols are a group of terpenoids with a chromanol ring structure and a 13-carbon side chain at the C2 position. This group includes tocopherols and tocotrienols, which are commonly known as vitamin E. Naturally, each of the tocols has four homolouges, i.e., α-, β-, γ-, and δ-, with the variation on the number and position of methyl groups attached to the chromanol ring (Figure 1). While tocopherols have three chiral carbons and a saturated side chain (*viz.* pythyl), tocotrienols contain a chiral carbon and an unsaturated side chain (*viz.* farnesyl) with three *trans*-configurations of carbon-carbon double bonds at C3′, C7′, and C11′ [1]. All of these properties lead tocols to a lipophilic compound (Table 1). Although they exhibit resistance to heat and stability under acidic conditions, tocols are unstable toward alkali, light, and oxygen [2]. The physicochemical characteristics of tocols are presented in Table 1. Furthermore, tocols exhibit physiological functions in the human body, such as neuroprotective [3,4], cardioprotective [5], inflammatory activity [6], cholesterol biosynthesis inhibition [7], antioxidant [8], anti-cancer with potent proapoptotic activity [9], anti-hypertensive [10], and glutamate-induced death preventive [11,12,13]. These attributes make them an attractive phytonutrient derived from oil-based plants.

Large scale production of tocols is performed either by catalytic synthesis or by extracting from natural sources [14,15]. However, the synthesis processes always produce a mixture of stereoisomers [16]. In the synthesis of α-tocopherol, for example, the route generates an equimolar mix of RRR-, SRR-, RSR-, RRS-, RSS-, SSR-, SRS-, and SSS-α-tocopherol; *viz.* all-racemic-α-tocopherol [17,18,19]. Meanwhile, natural tocols are exclusively found as single stereoisomers containing chiral carbons in the R-conformation [14,15,20], such as 2R, 4′R, 8′R-α-tocopherol [17,18] and 2R, 3′R, 7′R-α-tocotrienol [5,21]. The R- and other 2R stereoisomers exhibit higher affinity than the 2S stereoisomers [22].

As such, interest has increased in obtaining naturally occurring tocols compared with the synthetic ones nowadays [5] because of the isomer distribution of natural tocols to show more potent in biological function [23]. For example, natural α-tocopherol exhibits higher affinity with alpha-tocopherol transfer protein (α-TTP) than the synthetic one (1.36:1.0–1.49:1.1 bioavailability ratio) [18,24], leading to its faster absorption, transportation, and distribution in the human body. In the study of the rat resorption–gestation assay using doses ranging from 0.383 to 0.583 mg, the affinities of tocols on α-TPP were aligned with their activities in the order of β-tocopherols (38%) > α-tocotrienols (12%) > γ-tocopherols (9%) and > δ-tocopherols (2%) [25]. By using the fetal rat resorption model at doses of 2.5–10 mg/Kg body weight, the biopotencies of tocotrienols can be presented as α-tocotrienol (27.7 ± 9.2%), γ-tocotrienol (9.1 ± 2.4%), and δ-tocotrienol (8.5 ± 3.5%) [26,27].

**Table 1 molecules-30-01217-t001:** Physicochemical characteristics of tocols.

Tocols	Solubility (mg L^−1^) *	m.p. (°C) **	b.p. (°C) **	Log P	H-Bond Donor	Ref.
α-tocopherol	Miscible with chloroform, vegetable oils, ether, acetone and alcohol. Immiscible with water.	2.5–3.5	235.0	10.7	1	[28]
β-tocopherol	Chloroform (Sparingly), Ethanol (Slightly), Ethyl Acetate (Slightly), Methanol (Sparingly)	<25	474.9	10.3	1	[29]
γ-tocopherol	Chloroform (Sparingly), Ethanol (Slightly), Methanol (Sparingly)	<25	518.0	10.3	1	[30]
δ-tocopherol	Chloroform (Sparingly), Ethanol (Slightly, Sonicated), Ethyl Acetate (Slightly),	<25	464.7	10.0	1	[31]
α-tocotrienol	Chloroform (Slightly), Methanol (Slightly)	<25	541.7	9.3	1	[32]
β-tocotrienol	Soluble in ether, ethyl acetate, hexanes, and does not mix well water.	<25	507.5	8.9	1	[33]
γ-tocotrienol	Chloroform (Slightly), Ethyl Acetate (Slightly), Methanol (Slightly)	<25	530.8	8.9	1	[34]
δ-tocotrienol	Chloroform (Sparingly), Methanol (Slightly)	<25	541.7	8.6	1	[35]

* 25 °C, ** pressure 1 atm, m.p: melting point, b.p: boiling point, Log P: octanol–water partition coefficient.

Deodorizer distillates (DDs), derived from vegetable oil refining, are a potential source for producing tocols, considering their lower cost and higher concentration compared to vegetable oils. For example, the crude palm oil industry produce ~4% of palm fatty acid distillates [36], and about 15–57% of tocols are accumulated in the distillate fraction during oil refining process [14,15,37]. The type of deodorizer, quality of the input oil, and processing variables like vacuum, temperature, and deodorization duration can all affect the composition of the deodorizer distillate. The amount of DDs depends on the FFAs content of the input oil to the deodorization process rather than the type of oil used. However, the tocols content in DDs is highly dependent on the oil source used. If the main source for making the oil contains tocols, both tocopherols and tocotrienols, then it is likely that the DDs will contain tocols compounds. Based on the results of the literature study, the composition of various DDs that has been studied is summarized in Table 2, all of which contained tocopherol. However, only a few contain tocotrienols, particularly rice bran and palm. This is in accordance with previous research that tocopherols are synthesized by all organisms that perform photosynthesis, unlike tocotrienols, which are only found in several types of plants.

However, currently developed methods for the separation of tocols from DDs involve multiple steps [46], i.e., esterification, saponification, liquid–liquid extraction, crystallization, distillation [47], ion exchange, and adsorption [48], because tocols exist in a mixture with various structurally related compounds, especially fatty acids. In comparison, adsorption and ion exchange suffer from low capacity and high consumption of solvents while traditional organic solvents give low selectivity (20–28% extraction efficiency) [41,49,50]. A molecular distillation technique has been applied for tocols extraction from soya oil’s and rapeseed oil’s DDs with good recovery results, i.e., 35–50% (~35% purity of tocopherol) at 150–220 °C under vacuum (<5 Pa) [51]. However, its severe operation, i.e., high temperature and vacuum pressure, has triggered many attempts to explore other effective and efficient methods for selectively extracting tocols. Moreover, emerging environmental regulations to minimize the emissions of volatile and harmful organic solvents promote the application of green chemistry principles in all aspects of processes for sustainable production.

Currently, ionic liquids (ILs) are neoteric solvents that have attracted significant interest due to their distinct physical and chemical characteristics, such as non-flammability, lack of toxicity, low melting point, and minimal volatility [52,53]. These solvents have a strong ability to interact with organic molecules through various intermolecular bonds, e.g., μ−μ, dispersion, ionic exchange, and H-bonds [54]. In particular, molecules bearing −OH groups like phenols can act as H-bond donor (acidity), which can be effectively separated using appropriate neoteric solvents. Ren et al. [54] reported the selective separation of δ-tocopherol from their homologues (α-, β- and γ-) using a biphasic system of [bmim]-based ILs and hexane. A certain amount of mixed tocopherols was first dissolved in hexane. The solution was then mixed with an equal volume of the IL in methanol. The mixture was agitated for 3 h at 200 rpm and allowed to settle for at least 3 h. Extraction selectivity up to 21.3 and a distribution coefficient of 2.34 was attained for delta tocopherol over other tocopherol isomers using [bmim]Cl [54]. Ni, Xiaolei et al. (2012) presented the effectiveness of amino acid-based ILs for the selective extraction of α-tocopherol (29.6 selectivity, 92% recovery efficiency) from methyl linoleate as a model compound of DDs diluted in hexane. The study showed 29.6 selectivity and 92% recovery efficiency using a mixture of [emim]Ala and [emim]Lys as the extractant [49]. They used the same extraction method as Ren et al. [54]. In recent reports, Qin et al. [55,56] successfully extracted tocopherol homologs from soybean and corn oil deodorizer distillate with an imidazolium-based ionic liquid. In total, 1.00 g of methylated oil deodorizer distillates (MODD) was added using 1.92 g of ILs and stirred with a magnetic stirrer at 1000 rpm for a specific duration (between 0.5 and 3.0 h) at temperatures ranging from 25 °C to 55 °C. After settling for a period of 0.5 to 5 h, two distinct phases were observed in the vial. A recovery rate of 90–98% was attained by using [C_6_MIM]Acetate, with a selectivity of 108.23 and a partition coefficient of 8.182 for α-tocopherol. Imidazolium-based ILs have also been used for the extraction of tocotrienols from palm fatty acid distillate (PFAD) by Ulfa, D.M. et al. (2024), and it was found that 1-Butyl-3-methylimidazolium acetate had the highest efficiency of 75.41% [57].

Despite their effectiveness in tocols extraction, the practical application of ILs faces some issues resulting from their high viscosity, relatively high price, complex preparation, poor biocompatibility, and biodegradability [58]. Deep eutectic solvents (DESs) are another class of neoteric solvents that are promising alternatives to ILs. Besides having similar characteristics with ILs, DESs possess greener and more sustainable properties [59]. For instance, choline chloride (ChCl) is a cheap, safe, and biodegradable hydrogen bond acceptor (HBA) that can form DESs with numerous hydrogen bond donors (HBDs) derived from natural sources like urea, carboxylic acids, amino acids, and polyols [60]. An additional appealing aspect of DESs is their adaptability as reagents, allowing scientists to fine-tune their properties through the selection of suitable components and their respective ratios. Moreover, DESs can also be regenerated via antisolvent evaporation method [61]. These attractive features have prompted many studies to apply them in the extraction of natural compounds under mild condition, similar to ILs [62].

A number of variables, such as the ratio of HBA to HBD and extraction temperature, may influence the choice of DESs during the extraction process, thereby affecting yield [63]. In this review, the potential practical applications of DESs for the selective extraction of tocols (*viz*., tocopherols and tocotrienols) from DDs are discussed. Various kinds of DESs and their components are first presented in correlation with their effectiveness on the tocols extraction capacity and selectivity under different extraction techniques, especially, monophasic, in situ DESs formation, and biphasic processes. The intermolecular forces between the tocols and extractants were delivered and highlighted as important for obtaining the result with good extraction efficiency as well as selectivity. This variation varies depending on many factors, including the types of DESs components (H-bond donors, HBD and H-bond acceptors, HBA) and tocols’s isomers, which are related to the molecular structure configuration. As such, the information provided can guide the selection of an appropriate DESs system for extracting tocols from DDs selectively as well as effectively.

## 2. Tocols Extraction Using Deep Eutectic Solvents

Moving beyond organic solvents and ILs, DESs have emerged as another promising option for various chemical processes [64,65,66], including the extraction of tocols [67]. Typically, this solvent is a consistent liquid formed by easily combining at least two components, namely HBA and HBD that can form a single melting and freezing point lower than the melting point of its pure component [68,69]. Usually, a quaternary ammonium salt is used as the main component and it is mixed with metal halides, metal halide hydrates, or compounds acting as HBD. These compositions classify DESs as types 1, 2, and 3, respectively [62,70]. In addition, metal halide hydrates and HBD could also form DESs categorized as type 4 [71]. The intermolecular forces of different components of DESs form a homogeneous mixture with a lower melting point compared with their pure components. Specifically, H-bond interactions play a role in DESs. This interaction can be determined from the appearance of strong proton signals on the HBD and HBA in ^1^H NMR spectroscopy measurements (Figure 2). Moreover, the shifted O−H stretching vibration peak at 2800–3300 cm^−1^ and the carbonyl peak at 1700–1725 cm^−1^ in ChCl Fourier Transform Infrared (FTIR) spectra indicated the occurrence of intermolecular H-bonds in the formed ChCl-based DESs [72,73]. The use of DES for the extracting of tocols from various matrices has been steadily increasing as summarized in Table 3 from the deodorizer distillate and Table 4 for other sources.

### 2.1. Mono-Phasic Solvent System

The components of DESs has significantly influence their physicochemical properties, including their polarity and dissolving capacity. Liu et al. [43] reported the effectivenesses of tocopherol extraction from soybean oil deodorizer distillate (SODD) in several kinds of choline chloride (ChCl)-based DESs with organic acids (acetic acid, malonic acid), polyols (ethylene glycose, glycerol) or phenols (phenol, o-cresol, m-cresol, p-cresol). A specific quantity of SODD was added to a certain volume of DESs. The mixture was treated with 5 min vortex-assisted extraction and then centrifuged at 6000 rpm for 3 min until two phases were obtained at room temperature and atmospheric pressure. Tocopherol was recovered from DESs by adding water and hexane. DESs composed of ChCl and phenolics exhibited high efficiency (70–80%), with the activity of 6–8 folds higher than those of the other DESs mentioned above. Their ChCl-based DESs were also better than those prepared using methanol under the same condition. The worst performances of DESs prepared from ChCl and organic acids (malonic and citric acid) were also reported in tocopherol extraction from crude palm oil, in which the selectivity toward tocols homologues was less than half that obtained from ILs [C_6_MIM]Cl [73].

DESs composed of ChCl and phenolics (i.e., phenol, cresols, 4-methoxyphenol) have also been reported to be a good solvent for extracting lignans, e.g., sesamin, sesamolin, sesamol; in sesame oil. 0.2 g sesame oil diluted with n-hexane then added with DESs, subsequently placed in an ultrasonic bath to aid lignans extraction, followed by centrifugation for 5 min at 4000 rpm. 97–100% recovery was achieved by DESs composed of ChCl and phenolics, while such kind results were not observed from DESs composed of ChCl with polyols like while ethyleneglycol, propylene glycol, and glycerol [74]. Sesamin, sesamol, and sesamolin possess benzene rings similar to those of tocols. The delocalized *π*-*π* electronic in the benzene ring of those DESs’ component is presumed to be beneficial for additional interaction with substances bearing the phenol ring. It is supported from the result that it should be noted that the low efficiency of compared the high steric hindrance of the methyl group around the *π*-*π* electronic in the benzene ring of catechol contributed to the low efficiency of ChCl-catechol-based DES compared with ChCl-p-cresol-based DES, i.e., 125–750 mg kg^−1^ and 1000–2250 mg kg^−1^, respectively [74]. Therefore, the good effectivity of ChCl/phenolics-based DESs in extracting tocopherols from SODD under mild conditions indicates that the presence of the benzene ring of the phenolic plays a key role in achieving good interaction with tocopherol (Figure 3).

### 2.2. In Situ Dess Formation

Tocols possess a phenolic hydroxyl group in their molecular structure and weak HBD ability. This feature also leads them to play role as HBDs when they interac with organic salts bearing HBA, resulting in a eutectic mixture in situ. In this respect, Qin et al. [75] reported that the addition of tetrabutylammonium chloride [CH_3_(CH_2_)_3_]_4_NCl directly to SODD and corn oil deodorizer distillate (CODD) exhibited high extraction performances of tocopherols, i.e., >9.31 *β*-coefficient and >91% extraction ratio at 65 °C with [CH_3_(CH_2_)_3_]_4_NCl/*α*-tocopherol ratio of 20 and 2 h. Here, all isomers of tocopherols were extracted in the phase of eutectic mixture with [CH_3_(CH_2_)_3_]_4_NCl, which could be easily separated after extraction from the layers of [CH_3_(CH_2_)_3_]_4_NCl solely and DD. The formation of the eutectic was indicated by the higher solubility of [CH_3_(CH_2_)_3_]_4_NCl in CODD than in methyl linoleate and hexane (1.15, 0.15 and <0.01 g 100 g^−1^, respectively). However, the type of organic cation affects the successful in situ formation of the eutectic of tocols.

The selective extraction of tocols via in situ DESs formation from DD is a complex process. In the case of tocopherol extraction from SODD and CODD, Qin et al. [75] observed impurities like non-esterified fatty acids, sterol, and some small molecules to be also extracted into the eutectic phase tocopherols-[CH_3_(CH_2_)_3_]_4_NCl. DDs contain a mixture of compounds, particularly free fatty acids (non-esterified fatty acid and sterols). These compounds could also act as the HBD, and thereby, interacting with DESs and extracting together with tocols. Separating the eutectic phase and re-extracting it with an antisolvent like n-hexane, continuing with water as a disrupting agent were successfully achieved a concentrated tocols (~80%) from the mixture. The sequence addition of antisolvents as well as disrupting agent is required to maximally recover tocols from the mixture. As such, selective tocols extraction via in situ DESs formation can be achieved by separating the tocols-DESs extract phase continuing with cascade purification by adding antisolvents.

### 2.3. Bi-Phasic Solvent System

The multiple intermolecular interactions of DESs with tocols can separate later molecules from the matrix of the mixture and dissolve them in the DES phase. To obtain high-quality tocols, the valuable compounds from the DESs phase should be conducted in the next step. Crystallization is inappropriate due to the low melting point of tocols, which hinders their crystallization. Since DESs are low volatility compound, applying conventional evaporation at reduced pressure is also insufficient. Due to the polar nature of DESs, the use of low-polarity solvents such as hexane is mostly selected because tocols are relatively soluble in hexane [80], while DESs are completely immiscible with hexane (solubility 0.003%) [73,79]. In addition, this hydrocarbon has been shown as a good antisolvent for extracting all isomers of tocopherol from DESs (80–100% recovery) [43]. Moreover, the addition of water can improve the recovery because its strong polarity as well as HBD, e.g., α, *β*, *π** Kamlet Taft parameters of 1.23, 0.49, 1.14, respectively (Figure 4) [81], would distract the interaction between DESs and tocols [82], resulting in more effectiveness of the later molecules to dissolve in hexane [78].

Considering these phenomena and to improve the extraction efficiency, tocols extraction with DESs can be intensively optimized using a biphasic system with nonpolar solvents. The selection of the antisolvent is necessary to optimize the purification of tocols during biphasic extraction. Herein, the antisolvent should have the following criteria: interact strongly with DDs’s components but not significantly dissolve DESs as well as disrupt the tocols–DESs interaction. For instance, among different groups of organic solvents (i.e., esters, arenes, alkanes), hexane/[CH_3_(CH_2_)_3_]_4_NCl showed the best biphasic system to selectively extract tocopherol from methyl linoleate (70–91% extraction ratio) [67]. This is because alkanes like hexane, heptane, and octane are not good solvents for [CH_3_(CH_2_)_3_]_4_NCl due to their low polarity, while they interact strongly with methyl linoleate (Figure 4 and Figure 5A,B) [56,83]. A large peak arising of methyl linoleate in the nonpolar region showed its major distribution in hexane (Figure 5C). However, the small peaks in the region of HBA and the low negative value in the region of HBD but the high positive value in the region of HBA indicate the weak affinity of methyl linoleate for the interaction with HBD but is repulsive to HBA (Figure 5D). Therefore, methyl linoleate interacts to some extent with CH_3_(CH_2_)_3_]_4_NCl, similar to HBA. This result is supported by the fact that the *σ*-profile of methyl linoleate is similar to that of tocopherol (Figure 5).

## 3. Tocols—Deep Eutectic Solvents Interaction

### 3.1. Intrinsic Factors

Basically, liquid–liquid extraction of a solute depends on the polarity of its molecule and solvent. Typically, it is a function of multiple thermodynamic properties of the solute and solvent components. One such parameter is the *σ*-profile, which indicates charge-related and molecular-specific properties [84]. The *σ*-profile could be divided into three regions within different distributions of screening charge density: the nonpolar region (−0.0082 e/Å2 < *σ* < 0.0082 e/Å2), the HBA region (*σ* > 0.0082 e/Å2), and the HBD region (*σ* < −0.0082 e/Å2) [58]. In addition to indicating the polarity of individual molecules, this parameter could also exhibit the intermolecular affinity between different molecules. The intermolecular strength is responsible for the interaction of different components and, thus, determines the distribution of compounds in each solvent phase. The evaluation of this characteristic is primary significance for predicting and obtaining the potential interaction of tocols and solvents, particularly with DESs.

DDs mainly contain fatty acids and their derivatives in addition to tocols. The presence of these components affects the interaction of tocols with the solvent molecules. Figure 6A shows the *σ*-profile of *α*-tocopherol compared with methyl linoleate as a representative of the tocols molecule and fatty acid, respectively. In general, both tocopherol and methyl linoleate showed strong peak in the nonpolar region and weak peaks in the HBA region. Some representative fatty acids like hexadecanoic acid, cis-13-octadecanoic acid, *α*-linoleic acid, and triglycerides (glyceryl trilinoleate, trioleate, tripalmitate, tristearate) exhibit similar patterns of a strong peak *σ*-profile in the nonpolar region (Figure 6B–D) [85]. These findings clarify the similar molecular characteristics of tocols with some of the DD’s components and affirm the difficulties associated with their separation.

It should be noted that a small shoulder peak in the HBD region was observed in the *σ*-profile of *α*-tocopherol, indicating its weak HBD ability (Figure 6A). This feature is beneficial to allow tocols to interact more strongly with strong HBA molecules like DESs. In contrast, methyl linoleate does not exhibit peaks in this region. In the case of *α*-tocopherol extraction in methyl linoleate as a model of DDs, for example, methyl linoleate was extracted to be only ~9% in DES phases of CH_3_(CH_2_)_3_]_4_NCl [75]. Interestingly, *α*-linolenic acid (i.e., an acid form of methyl linoleate) and organic acids, including common fatty acids observed in plant oils exhibit similar small peaks in the HBD regions of their *σ*-profiles as well (Figure 6D). Similarly to methyl linoleate, such small peaks are absent in the triglyceride form of fatty acids (Figure 6B). These phenomena indicate that fatty acids could also interact with DESs during tocols extraction from DDs, leading some of them to be extracted and might decrease the obtained tocols. Therefore, the tocols extraction performance from DDs was lower than that obtained from alkyl esters as a model deodorizer distillate. For instance, the recovery ratio of tocopherol extraction from methyl linoleate was reported to be 92% with >99% purity, whereas it took 85–92% recovery ratio with 80% purity from SODD and CODD [75]. In fact, the total extraction of tocotrienol and tocopherol from crude palm oil (CPO) using ChCl-based DESs of organic acids (acetic, oxalic, citric) showed higher efficiencies than those obtained from CPO-diacylglyceride and CPO-monoacyglyceride, i.e., 2531–4439, 1397–3539, and 488–627 mg kg^−1^, respectively [72].

The tocols interaction with an HBA varies depending on many factors, e.g., the types of HBA. Figure 7A presents the predicted infinite extraction capacity and selectivity of several organic salts that are involved in DESs, ILs, and organic compounds (non-salts) with *α*-tocopherol. In general, alkylammonium-based salts exhibit better interaction with *α*-tocopherol than imidazolium-, pyridinium-, phosphonium-, and cholinium-based salts. Moreover, they have the worst affinity with all organic compounds non-salts such as diols, organic acids, sugars, amino acids, and amines. Here, [CH_3_(CH_2_)_3_]_4_NCl demonstrated the best performance, whereas cholinium salts like ChCl lack affinity. This is in line with the experimental results reported by Qin et al. [75], in which a homogenous in situ DES of [CH_3_(CH_2_)_3_]_4_NCl and *α*-tocopherol was successfully formed, whereas such a homogenous solution was not formed when *α*-tocopherol was mixed with ChCl or [CH_3_CH_2_]_4_NCl. One point of view is that the intermolecular bonding capability, i.e., hydrogen (H-), van der Waals, and misfit bonds of organic salts could determine their significances in their interaction with tocols. In the case of [CH_3_(CH_2_)_3_]_4_NCl, [CH_3_CH_2_]_4_NCl and ChCl, the former salt has the tightest interaction with *α*-tocopherol, while ChCl has the weakest interaction (Figure 7B).

The strengthening of intermolecular bonds is essential for the effectiveness of molecule–molecule interactions. Usually, the molecular structure of a compound affects the stereochemistry arrangement of its atoms and the strength of intramolecular forces (i.e., ionic-, covalent-, dipole–dipole bonds) [87]. These variations lead to different interaction with the surroundings. For instance, [CH_3_(CH_2_)_3_]_4_NCl possesses the weakest ionic bond energy (ΔE) (−364.369 kJ mol^−1^) and longest length (3.92535 Å) of [CH_3_(CH_2_)_3_]_4_N^+^ cation and Cl^−^ anion compared with [CH_3_CH_2_]_4_NCl (ΔE = −367.212 kJ mol^−1^, 3.89233 Å) and ChCl (ΔE = −412.075 kJ mol^−1^, 3.89233 Å), allowing the Cl^−^ anion of the former salt to be more electronegative and free to contact with *α*-tocopherol, while ChCl is the less active one [75]. The stronger interaction of alkylammonium salts with *α*-tocopherol with ILs likely also resulted from the weak bonds of their cations and anions compared with the ionic bonds in ILs, which gives the polarizability charge of the ions of alkylammonium salt to be dense.

The steric hindrances of the molecules may also affect the selectivity of the extraction. The extraction of tocopherols from CODD using [CH_3_(CH_2_)_3_]_4_NCl showed different extraction coefficients and recovery ratios for each tocopherol isomers in the order of *α*-tocopherol (*β* = 9.3, *η* = 81.3%) < *β*-/*γ*-tocopherol (*β* = 14.2, *η* = 91.2%) < *δ*-tocopherol (*β* = 15.8, *η* = 92.4%) [75]. Meanwhile, the selectivity of DESs made from ChCl and sucrose (2:1 molar ratio) toward tocopherols is *α*-tocopherol > *β*-tocopherol > *γ*-tocopherol > *δ* tocopherol [79]. Since the chromanol ring is the main active site of tocols as a HBD [43], the presence of methyl groups surrounding the hydroxyl group has various effects. First, it gives high steric hindrance to the chromanol ring of tocopherol isomers, which affects the interaction of each isomer with HBA similar to DESs (Figure 8). Second, the methyl groups contribute to the distribution of electronic potential on the molecule surface, especially around the *π*-*π* electron of the benzene ring [79,88]. These properties affect the extraction efficiency and selectivity of the tocols isomers [75]. This is supported by the comparative performances of *β*- and *γ*-tocopherol extraction because of their similar methyl groups. The atomic configuration on a DES’s molecules or HBD has similar effects on the interaction with the tocols.

Polarity is one of the most significant intrinsic characteristics of DESs because it serves as a primary indicator of the ability of DESs to dissolve and potentially interact with others [93]. In fact, different tocols have different polarities, e.g., α-tocopherol < α-tocotrienol < β-tocotrienol < γ-tocotrienol < δ-tocotrienol [83]. Herein, the Kamlet–Taft parameters are commonly used to describe it in association to ascertain the likely factors affecting their extractability. The parameters include α, β, and π*, which represent HBD, HBA and, dipole/polarizability, respectively. Therefore, the Kamlet–Taft polarity parameters of the DESs were useful for elucidating the likely factors affecting the extractability of DESs. Table 3 presents the results for the α (donating hydrogen bonds), β (accepting hydrogen bonds), and π* (dipole/polariz ability) [94]. Tocols are insoluble in water due to the lengthy tail alkyl groups in the structure (refer to Figure 1). Additionally, because tocols also contain one phenolic hydroxyl group, tocols and DESs can interact through hydrogen bonds, making DESs an effective solvent for dissolving tocols. With the highest α × β value and the highest hydrogen bonding ability, it is expected that DES ChCl-LA will have the highest extraction efficiency, and the hydrogen bonding ability can be represented by the α × β value [95].

### 3.2. Extrinsic Factors

In contrast with plant oils, the direct use of DDs faces the issue of their semi-solid form at room temperature due to their high free fatty acids content as a major compound [88,90]. Generally, dissolution of this oil-refining side product into DESs at room temperature is difficult because the high viscosity of DESs limits their mass transfer. In particular, it is ineffective for some of DESs in the solid form at room temperature. Applying mechanical stirring was reported to be able to improve mass transfer rate, although this requires a time to achieve homogenization well compared with giving ultrasonic [41].

Since DDs melt at around 32–38 °C and DESs possess good stability at <100 °C, heating the mixture during extraction is the most efficient way because it can lower the DESs’ viscosity and melt the DDs. [41]. For example, performing extraction at 55 °C can give a well mixture of SODD and a DES of ChCl/p-cresol without reducing the tocopherol extraction efficiency [41,74]. However, the temperature selection was dependent on the characteristic of DESs. In the case of DESs of ChCl-oxalic acid and ChCl-citric acid, these temperatures are not appropriate since they are starting to melt at 90 and 170 °C, respectively, whereas it is suitable for ChCl-acetic acid with a melting point of 10 °C [71]. However, it should be noted that tocols are degraded at 210 °C [91].

Liu et al. [41] reported that extraction efficiency of tocopherol can be increased to some extent (~1.7-folds) by enhancing the ratio of ChCl/p-cresol DES and DDs [41]. By using methyl linoleate as a model, the selectivity of α-tocopherol extraction was doubled by increasing the mass ratio of [CH_3_(CH_2_)_3_]_4_PCl/ethanolamine and methyl linoleate from 0.5 to 1.5 [58]. Similar trends have been reported for tocopherols or tocotrienol extraction from crude palm oil with ChCl-based DESs [71,72,74]. These results demonstrate that increasing the ratios is beneficial to improve the potential interaction of DES molecules with tocols, thereby benefiting the extraction performance. Nevertheless, the magnitude effect varied for different tocols’s isomers depending on the types of DESs, which are related with the variety of DESs-tocols interaction as discussed above.

## 4. Toxicity of DESs

Despite its numerous benefits, the field of DESs research is still relatively growing, with ongoing research focused on examining their characteristics, understanding their functionality, and enhancing the design of these entities for practical application. Because of their low volatility and low toxicity, they are frequently viewed as more environmentally friendly and sustainable [92,93]. A comprehensive examination is required to confirm whether DESs are truly “green” as argued by Wen et al. (2015a) [94].

Research has been conducted to assess the toxicity of DES [96,97,98] with the aim of investigating its impact on the environment and living organisms [99]. Research has focused on genotoxicity, ecotoxicity, acute and chronic toxicity, and biodegradability. The results indicates that certain formulations of DESs could be more environmentally friendly and pose a lower risk than conventional organic solvents.

Several chlorocholine-based deep eutectic solvents have been shown to exhibit cytotoxic profiles on fibroblast cells, with their toxic effects correlated and quantified in relation to their viscosity [100]. Research findings indicate that high viscosity in DES hinders shrimp movement, and oxygen deficiency is another possible contributing factor, which may result in adverse effects on it [101]. Recent findings indicate that DES can display significant synergistic effects, and its cytotoxicity levels were found to be greater than those of its individual components, particularly in DES formulations based on choline chloride (ChCl). The degree of toxicity was also found to be influenced by the type of organism in question. Type 1 DESs formed from metal and quaternary ammonium salts and type 2 DESs consist of hydrated metal halides, and choline chloride are more toxic than type 3 DESs formed from choline chloride and hydrogen bond donors, primarily due to their composition which includes metal salts in addition to organic salts, in contrast to type 3 DESs, which are comprised solely of organic salts. Unlike ILs, the toxicity level of DESs is influenced by the specific individual salt and its HBD, with stronger hydrogen bonds resulting in lower toxicity [102]. A summary of all human and animal cells used for testing the toxicity of DESs is also presented in Table 5.

## 5. Summary and Outlooks

Tocols or vitamin E are naturally lipophilic and comprise two classes: tocopherols and tocotrienols. Although they could be commercially produced via synthesis, the current growing market demand for natural-based tocols has attracted much interest due to the greater potential health benefit of the natural tocols than synthetic tocols. Plant oils are the main source of tocols; however, the use of deodorizer distillates (DDs) from the vegetable oil refineries has the potential to increase the economic value of oil refinery streamlines and achieve sustainable development goals. Current existing technology for tocols extraction from DDs applies molecular distillation technique to yield 35–50% (~35% purity) of tocopherol; however, harsh operating conditions (150–220 °C, <5 Pa) have led to many attempts to find other techniques with the main goal of selective extraction with more moderate operations. Since most DDs comprise fatty acids (~80%), liquid–liquid extraction of tocols using conventional nonpolar organic solvents is challenging because of the similar nonpolarity of tocols with the DD’s components.

Naturally, tocols contain a long-chain hydrocarbon with a chromanol ring containing a hydroxyl group that could act as a weak HBD. This feature opens a way to utilize an HBA molecule for interacting with tocols via intermolecular forces, as exhibited by the effectiveness of ILs in extracting tocols at a mild temperature (>90% recovery at <50 °C). However, the application of ILs on a large scale is impractical considering the multiple drawbacks of ILs, such as poor biocompatibility and expensive and complex preparation [58,59,89]. DESs show promising options as an extracting agent due to their comparable characteristics and performances with ILs with superior properties, e.g., being biodegradable, less toxic, and affordable. Multivariation of DESs could be easily prepared from quaternary ammonium salt with the compounds bearing organic acids, polyols, and aromatics, even by using naturally derived compounds like amino acids. DESs composed of aromatics like ChCl-phenols exhibit effective extraction (70–80% efficiency) in a mild condition (room temperature), with activity being 6–8 folds higher than ChCl-based DESs with organic acids (acetic acid, malonic acid) and polyols (ethylene glycose, glycerol). The DES component effectively interacts with tocols via intermolecular forces (van der Walls, H-Bond, dipol–dipol), leading tocols to be efficiently separated from DD mixture components. In particular, the intermolecular H-bonds and the delocalized π-π electronic structure in the benzene ring could maximalize the interaction between tocols and DESs bearing phenolic groups.

In addition, some quaternary alkylammonium chlorides like [CH_3_(CH_2_)_3_]_4_NCl can strongly interact with tocols, generating an in situ DES that is easily separated from a mixture of DDs. These phenomena are not significantly observed for most quaternary ammonium salts like imidazolium-, pyridinium-, phosphonium, and cholinium-based salts. The molecular structure of quaternary alkylammonium salts plays an important role in providing significant multiple intermolecular bonding capability, i.e., hydrogen (H-), van der Waals, and misfit bond, for interacting with tocols. In the case of [CH_3_(CH_2_)_3_]_4_NCl, [CH_3_CH_2_]_4_NCl, and ChCl, the former salt accounts for the tightest interaction with α-tocopherol, while ChCl has the weakest interaction because [CH_3_(CH_2_)_3_]_4_NCl possesses the weakest ionic bond energy (ΔE) (−364.369 kJ mol^−1^) and the longest length (3.92535 Å) of [CH_3_(CH_2_)_3_]_4_N^+^ cation and Cl^−^ anion compared with [CH_3_CH_2_]_4_NCl (ΔE = −367.212 kJ mol^−1^, 3.89233 Å) and ChCl (ΔE = −412.075 kJ mol^−1^, 3.89233 Å). As a result, the Cl^−^ anion of the former salt is more electronegative and allows contact with α-tocopherol while ChCl becomes the less active. The stronger interaction of alkylammonium salts with α-tocopherol than ILs should also result from the weak bonds of their cations and anions compared with the ionic bonds in ILs, which gives the polarizability charge of the ions of alkylammonium salt to be dense.

In summary, the effective extraction of tocols with DESs as an HBA is affected by the following: (1) the larger fractional free volume, which results in more available free volume inside the ion cluster to allow more efficient HBD–HBA interaction to form a eutectic mixture; (2) the stronger intermolecular bonds (H-bond, van der Walls bond, misfit interaction) with tocopherol; (3) the lowest interaction Gaussian energy ΔE between cation and anion, which allows the anion of organic salt to be more free on interacting with tocopherol [89]. These characteristics should be strictly considered to obtain an appropriate DESs system for obtaining good extraction performances. Furthermore, mechanical stirring and heating during the extraction are essential to overcome the mass as well as energy transfer limitation due to the semisolid physical state of DDs. The major component of DDs is fatty acids that also have HBD ability, and, thus, the interaction of DESs with tocols would compete with fatty acids. One plausible strategy is the conversion of fatty acids into their glyceride form. In this context, esterification of fatty acids into their ester form would modify their potential interaction with DESs, result in loss of their HBD ability, and un-significantly disrupt tocols–DESs interactions. However, the reaction should be strictly controlled to hinder tocols degradation by a heat, acid-/base-catalytic reactions and oxidation. However, esterification for fatty acid removal should be strictly controlled because of the sensitiveness of tocols towards alkaline conditions, heat, and oxidation [105]. The simplicity and mild operating conditions (low temperature (<100 °C), ambient pressure) exhibit potential application of DESs for the direct extraction of tocols from DD. Moreover, considering the intrinsic properties of DESs that possess multiple interaction capability might be useful for the selective extraction of natural-based phenolic components.

Attention has been focused on the extraction of tocols from deodorizer distillate using deep eutectic solvents (DESs), largely due to the potential benefits it offers over traditional organic solvents. Evaluating the economic feasibility of scaling up this extraction method for commercial use involves considering factors such as cost assessment, process optimization, and environmental effects. Cost of raw materials: the distillate from the deodorization step in edible oil refining serves as a byproduct, often being more affordable than the raw materials used in the production process. Using this waste product has two main benefits: it decreases the cost of raw materials and gives value to previously underutilized resources. DESs are normally made of cheap and easily accessible components, like choline chloride and various hydrogen bond donors (e.g., urea, glycerol). The cost of these solvents is generally lower than that of conventional organic solvents, making them an economically attractive option for large-scale extraction. Optimizing production procedures: Research indicates DESs are capable of obtaining high extraction rates of tocols from the deodorizer distillate. The extraction process’s efficiency has a direct impact on the overall economic viability, because increased extraction yields result in lower costs per unit of extracted vitamin E. The DES extraction process typically requires less energy than conventional techniques, including Soxhlet extraction and supercritical fluid extraction. Reduced energy consumption translates into lower operational costs, enhancing the economic viability of scaling up the process. Environmental impact: The DESs are consistent with green chemistry principles, given that they break down easily in the environment and are harmless to health. Reducing the environmental impact is a growing concern for businesses seeking to minimize their ecological footprint. The potential for lower waste generation and reduced environmental impact can lead to cost savings in waste management and regulatory compliance. Tighter environmental regulations can make compliance more expensive, but shifting to environmentally friendly extraction methods could offset these costs by reducing potential fines and penalties, thus, making the process more financially viable. More specifically, technoeconomic research on the extraction of tocols from deodorizer distillates using deep eutectic solvents is urgently needed.

Deep eutectic solvents are increasingly being acknowledged for their notable reusability and stability in extraction procedures. Unlike conventional organic solvents, which frequently deteriorate or lose potency following a single application, DESs can be reused numerous times without substantial degradation of the extraction effectiveness. This characteristic not only helps reduce costs in industrial settings but also supports environmentally friendly practices by lowering solvent waste levels. DESs demonstrate exceptional resistance to thermal and chemical degradation, enabling them to maintain their integrity under a range of extraction conditions. Research has shown that DESs retain their solvent properties and extraction capabilities over multiple cycles, making them a desirable choice for industrial-scale applications. Effectively recycling DESs boosts their economic feasibility and environmental durability, thereby making them a more desirable option for extracting valuable compounds, such as tocopherols, from sources like deodorizer distillates. Extraction with DES has several notable benefits, but studies on DES toxicity reveal that certain deep eutectic solvents possess toxicity, although only a limited number of studies have been conducted to date. Ongoing toxicity evaluations are recommended for these eco-friendly solvents.

## Figures and Tables

**Figure 1 molecules-30-01217-f001:**
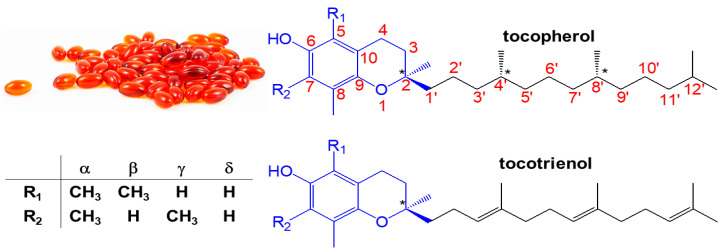
Basic structures of tocopherols and tocotrienols; the chromanol ring and pythyl/farnesyl side chain are shown in blue and black colors, respectively; the chiral carbon is denoted by an asterix (*).

**Figure 2 molecules-30-01217-f002:**
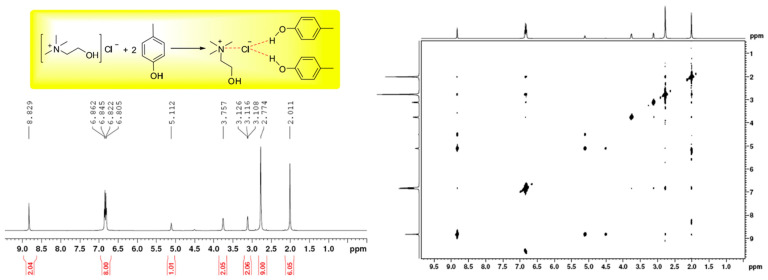
Intermolecular H-bonds of ChCl and *p*-cresol observed in its DES form at 1:2 molar ratios by ^1^H-NMR spectroscopy (**left**) and its nuclear Overhauser effect (NOESY) (**right**). The 2D NOESY exhibits a strong interaction of the protons on the hydroxyl group of ChCl and *p*-cresol. (Adapted with permission from [74]).

**Figure 3 molecules-30-01217-f003:**
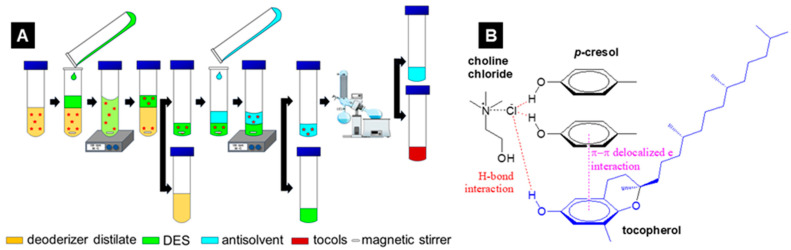
Illustration scheme of tocopherol extraction from Deodorizer Distillates using ChCl/p-cresol Deep Eutectic Solvent (**A**) and the intermolecular bonds of Deep Eutectic Solvents with δ-tocopherol through high π-π interaction (**B**).

**Figure 4 molecules-30-01217-f004:**
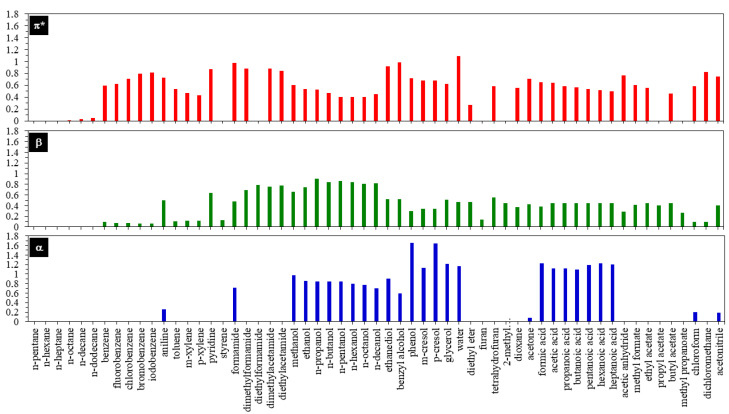
Kamlet Taft parameters of some organic compounds. *α*, *β* and *π** represent Hidrogen Bond Donor (HBD) ability, Hidrogen Bond Acceptor (HBA) ability and polarity/polarizability, respectively. (Adapted with permission from Ref. [83]).

**Figure 5 molecules-30-01217-f005:**
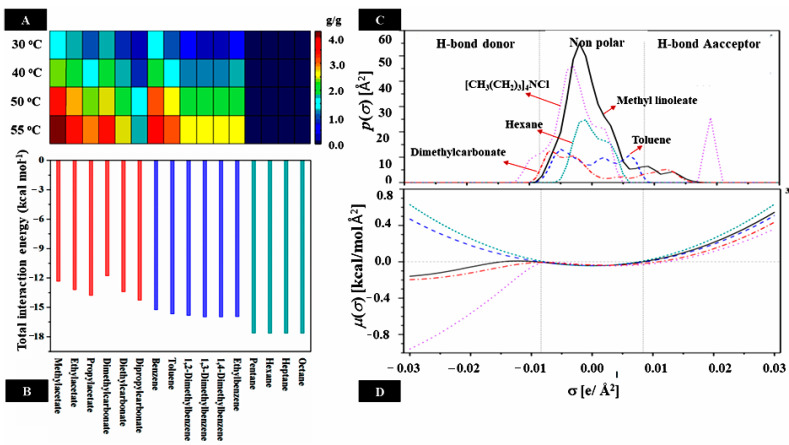
Some thermodynamic characteristics of tocopherol extraction in methyl linoleate with [CH_3_(CH_2_)_3_]_4_NCl and antisolvents predicted by COSMO-RS. (**A**) Solubility of [CH_3_(CH_2_)_3_]_4_NCl in several antisolvents. (**B**) Total interaction energy (misfit, van der Waals) of antisolvents and methyl linoleate. (**C**,**D**) *σ*-profile and *σ*-potential of some components. (Adapted with permission from Ref. [67]).

**Figure 6 molecules-30-01217-f006:**
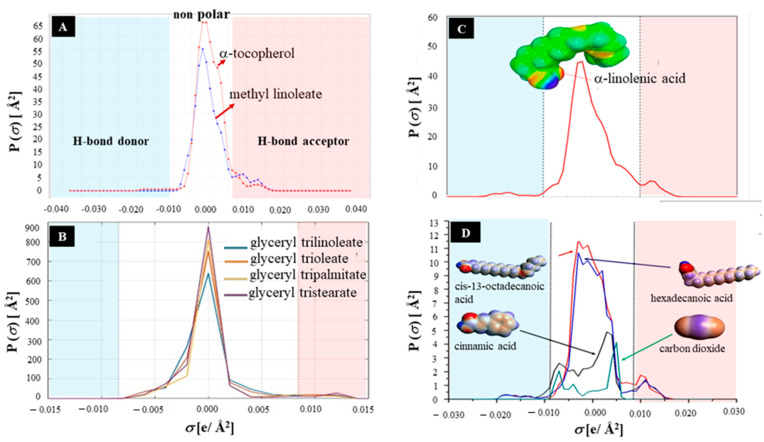
The σ-profile of α-tocopherol (**A**) compared with some organic compounds (**B**–**D**). (Adapted with permission from Refs. [85,86]).

**Figure 7 molecules-30-01217-f007:**
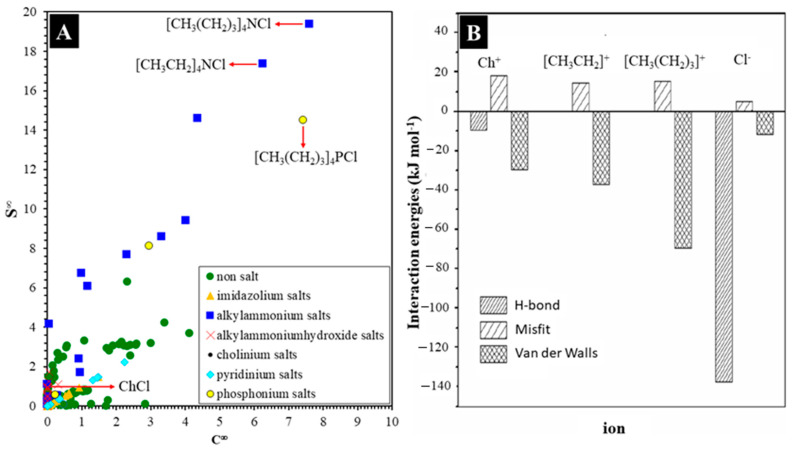
The selectivities and extraction capacities of some organic compounds (**A**) and the intermolecular bonds of some cations of organic salts (**B**) with tocopherol (Adapted with permission from Refs. [58,75]).

**Figure 8 molecules-30-01217-f008:**
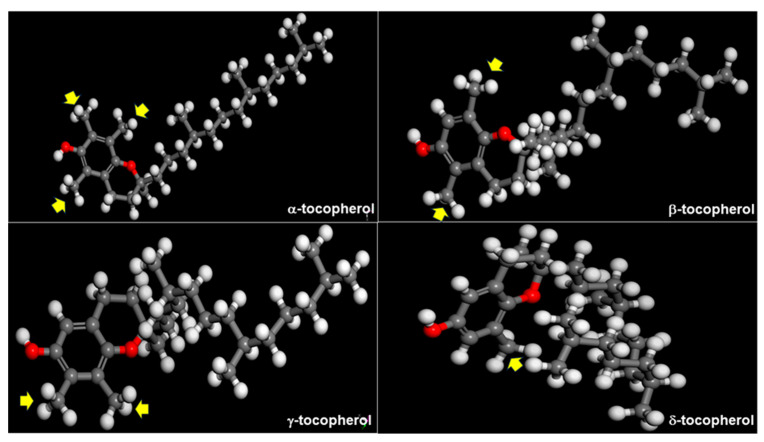
Molecular structures of *α*-tocopherol [89], *β*-tocopherol [90], *γ*-tocopherol [91], and *δ*-tocopherol [92]. Methyl group of chromanol ring is shown by arrow.

**Table 2 molecules-30-01217-t002:** Tocols content in different deodorizer distillate sources.

Oil Source [References]	Tocopherols				Tocotrienols			
	α-	β-	γ-	δ-	α-	β-	γ-	δ-
Rice Bran [38]	1.95	0.19	2.58	1.02	1.56	nd	4.46	0.11
Palm [39]	0.8	nd	nd	nd	0.19	nd	0.49	0.06
Canola [40]	1.44	nd	3.88	2.78	nd	nd	nd	nd
Rapeseed [41]	5.50	1.00	6.40	1.70	nd	nd	nd	nd
Sunflower [42]	17.08	nd	29.18	16.20	nd	nd	nd	nd
Soybean [43]	0.83	nd	6.84	5.69	nd	nd	nd	nd
Olive [44]	1.54	nd	1.67	0.03	nd	nd	nd	nd
Corn [45]	0.15	0.06	1.09	0.12	nd	nd	nd	nd

Values are in mg 100 g^−1^ unit, nd: not detected.

**Table 3 molecules-30-01217-t003:** Extraction of tocols from deodorizer distillate using different types of deep eutectic solvents.

Type	Analytes	DESs	Method of Extraction	Result	Ref.
SODD	α, γ and δ-T	ChCl: Acetic acid/Malonic acid/Ethylene glycol/Glycerol/Phenol/o/m/p-cresol	DES Preparation: heated and stirred at 60 °C Extraction: vortex for 5 min then centrifuged at 6000 rpm for 3 min, RT Recovery Tocol: n-hexane and water	extraction efficiency 77.6%	[43]
SODD and CODD	α-T	[N4,4,4,4]Cl	Extraction: stirrer at 1000 rpm/min, 2 h, 65 °C Recovery Tocol: hexane and water	Cβ-/γ-, and δ-tocopherol SODD (2.65% and 2.48%)	[75]
				CODD (0.84% and 0.71%)	
MODD	α-T	([N4,4,4,4]Cl	Extraction: stirred at 1000 rpm for 3 h, 55 °C, mixture settled for another 3 h at 35 °C	extraction ratio of α-tocopherol 91.2%	[38]
MODD SODD	α-T α, γ, δ -T	tetrabutylphosphonium chloride−ethanolamine (2:1) 12 kind TBAC-DES Based	DES Preparation: vigorously stirred for 3 h at 60−80 °C Extraction: stirred for 3 h, settled for another 3 h, RT Extraction: Sample dissolved in hexana add DES, vortex 5 min, centrifugated 4000 rpm 5 min,	β of 2.43 and S of 9.36 Max Extraction efficiency α, γ, δ -T: 85.0; 99.1; 98.0, Total Tocol: 97.5%	[58] [76]

Soybean oil deodorizer distillate (SODD), corn oil deodorizer distillate (CODD), methylated oil deodorizer distillate (MODD), tetrabutyl ammonium chloride (TBAC), tocopherol (T), tocotrienol (T3), cholin chloride (ChCl), room temperature (RT), concentration (C), selectivity (S), distribution coefficient (β).

**Table 4 molecules-30-01217-t004:** Extraction of tocols from sources other than deodorizer distillate using different types of deep eutectic solvents.

Matrix	Analytes	DESs	Method of Extraction	Result	Ref.
Red Palm Biodiesel	T and T3	K2CO3: glycerol (1:5, 1:6 and 1:7)	Preparation: DES in methanol) mixed with Biodiesel in hexane Extraction: 400 rpm for 3, RT Recovery Tocol: water-hexane mixture (4:1, *v*/*v*)	C_tokol_ 801.23 ppm	[77]
Crude Palm Oil (CPO)	T and T3	ChCl: acetic acid (1:2) ChCl: malonic acid (1:1) ChCl: citric acid (3:2)	Preparation: DES in methanol) mixed with CPO in hexane (1/1–5/1) Extraction: 200 rpm, 25 °C for 3 h Recovery Tocol: water-hexane mixture (4:1, *v*/*v*)	Cextract 8671 mg/kg, Control 3285 mg/kg)	[78]
Crude Palm Oil (CPO)	T and T3	ChCl and acetic acid glacial (1:2) ChCl and oxalic acid (1:2) ChCl and citric acid (1:2)	Preparation: DES in ethanol) mixed with CPO in hexane Extraction: 200 rpm, for 3 h, RT Recovery Tocol: water-hexane mixture (1:1, *v*/*v*)	C_tokol_ 4439 mg/kg extraction efficiency 74.98%	[72]
Crude Palm Oil (CPO)	T and T3	ChCl and malonic acid (1:1) choline chloride and citric acid (1.5:1)	Preparation: DES in methanol) mixed with CPO in hexane (2/1–5/1) Extraction: 200 rpm, 25 °C for 3 h Recovery Tocol: water-hexane mixture (4:1, *v*/*v*)	Distribution coefficients for α-T, -, β, γ-, δ and δ-T3: 7.8, 13.1, 19.8, 22.1 and 29.6	[73]
Ternary mixtures of {n-hexane tocopherol}	T	ChCl and (mono-, di- and tri-ethylene glycol/triethanolamine/sucrose (1:5)	Extraction: 550 rpm for 12 h, 25–35 °C	β (0.5118) and S (1.1679) for sucrose	[79]

Tocopherol (T), tocotrienol (T3), selectivity (S), distribution coefficient (β).

**Table 5 molecules-30-01217-t005:** Toxicity test of deep eutectic sSolvents with animals and human cells.

No	HBA	HBD	Cells/Species	Level Toxicity	Ref
1	Choline Chloride	glucose, glycerol, and oxalic acid	fish and human cell line	chloride:oxalic acid moderate cytotoxicity (EC50: 1.64 mM and 4.19)	[103]
2	Cholinium Chloride	acetic, citric, lactic, and glycolic acids	marine bacteria V. f ischeri	intermediate toxicity	[104]
3	Ammonium	glycerine (Gl), ethylene glycol (EG), triethylene glycol (TEG) and urea (U)	In vitro: OKF6, MCF-7, A375, HT29, and H413 In vivo: ICR mice	DES did not cause DNA damage, but it could enhance ROS production and induce apoptosis in treated cancer cells	[105]
4	Choline Chloride	malic acid, citric acid, lactic acid, fructose, xylose, mannose	Channel Catfish Ovary (CCO) cell line	low cytotoxicity	[96]
5	Polyethylene Glycol	lactic axid, Propanoic acid, urea, acetamide	lung cancer cell (A549)	DESs are more toxic than their individual components	[97]
6	Choline chloride, Betaine, Citric acid	oxalic acid, urea, xylitol, sorbitol, glucose, proline	HeLa, MCF-7, and HEK293T	formation of calcium oxalate crystals inside the cells induced detrimental effects on both tumor and normal cells	[98]
7	[Chol]Cl, [N1111]Cl, and [N4444]Cl	hexanoic and butanoic acid, ethylene glycol, 1-propanol and urea	keratinocytes (HaCaT) and tumor melanocytes (MNT-1)	[N4444]Cl-based DES, showed cytotoxicity for both cell lines	[99]
8	ChCl and N,N-diethylammonium chloride (DAC),	urea, glycerol, ethylene glycol, malonic acid and zinc chloride	HelaS3, AGS, MCF-7, and WRL-68	ChCl-based DESs (279 ≤ IC50 ≥ 1260 mM) were less toxic than DAC-based DESs (37 ≤ IC50 ≥ 109 mM)	[100]
9	ChCl and Betaine	sucrose, 1,4-butanediol, xylitol (2:1), 1,2-propanediol, Fructose	Caco-2, HeLa and HepG2 cells and peripheral blood mononuclear cells (PBMCs)	DESs at concentrations below 1%, affected tumor cells; however, healthy PBMCs were unaffected	[101]

## Data Availability

Data are contained within the article.

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
