# Peer review of "New Horizon in Selective Tocols Extraction from Deodorizer Distillates Under Mild Conditions by Using Deep Eutectic Solvents"

_molecules, 2025, doi:10.3390/molecules30061217_

Round 1

Reviewer 1 Report

Comments and Suggestions for Authors

The review article “New Horizon in Selective Tocols Extraction from Deodorizer Distillates Under Mild Conditions by using Deep Eutectic  Solvents”  presents a novel approach to selective tocol extraction using DESs. It is well-structured, with valuable findings; however, significant English language improvements are needed for better readability and clarity. Many sentences are grammatically incorrect, overly complex, or awkwardly phrased, making it difficult to follow the scientific discussion. Additionally, there are inconsistencies in terminology, verb tense, and article usage throughout the text. I have pointed out a few instances for improvement in the English language below along with other comments. However, thorough proofreading is required for the whole manuscript.

  1. L30, Abstract, remove in and with. It should be “in tocols extraction is presented considering three different ways”.
  2. L 38, Change to “phenolic structure and also briefly discusses the toxicity of DES.”
  3. L44-45, change to “that contain a” and replace on with “in their structure”.
  4. L58, change to "These attributes make them”
  5. L75, change to For example
  6. Table 1, in the footnote, expand the full form of Mr.
  7. L99-98, change to “Deodorizer distillates (DDs), derived from vegetable oil refining, are a potential source for producing tocols, considering their lower cost and higher concentration compared to vegetable oils.”
  8. Table 2 title, change to “Tocols content in”
  9. Be consistent with the format of et al in the manuscript, in some places, it is italicized and in some places it is normal.
  10. Section 2, avoid using abbreviations in section headings
  11. Figure 2 caption, change DESs to DES and close the bracket at the end of the caption.
  12. Table 3 title, change to Extraction of Tocols from Deodorizer Distillate using different types of DESs.
  13. Change the title of Table 4, same as Table 3
  14. Avoid using abbreviations in Table titles and figure legends
  15. L228, revise, it is incorrect.
  16. L341, change to “could be divided”
  17. Authors should explain Type I, II, and III DESs.
  18. Authors should include a Table on comparative analysis of extraction with conventional solvents or methods.
  19. Authors should include a discussion on the economic feasibility of scaling up this method for industrial applications.
  20. A paragraph on the reusability and stability of DESs in extraction processes should be added.
Comments on the Quality of English Language

The review article “New Horizon in Selective Tocols Extraction from Deodorizer Distillates Under Mild Conditions by using Deep Eutectic  Solvents”  presents a novel approach to selective tocol extraction using DESs. It is well-structured, with valuable findings; however, significant English language improvements are needed for better readability and clarity. Many sentences are grammatically incorrect, overly complex, or awkwardly phrased, making it difficult to follow the scientific discussion. Additionally, there are inconsistencies in terminology, verb tense, and article usage throughout the text. I have pointed out a few instances for improvement in the English language below along with other comments. However, thorough proofreading is required for the whole manuscript.

  1. L30, Abstract, remove in and with. It should be “in tocols extraction is presented considering three different ways”.
  2. L 38, Change to “phenolic structure and also briefly discusses the toxicity of DES.”
  3. L44-45, change to “that contain a” and replace on with “in their structure”.
  4. L58, change to "These attributes make them”
  5. L75, change to For example
  6. Table 1, in the footnote, expand the full form of Mr.
  7. L99-98, change to “Deodorizer distillates (DDs), derived from vegetable oil refining, are a potential source for producing tocols, considering their lower cost and higher concentration compared to vegetable oils.”
  8. Table 2 title, change to “Tocols content in”
  9. Be consistent with the format of et al in the manuscript, in some places, it is italicized and in some places it is normal.
  10. Section 2, avoid using abbreviations in section headings
  11. Figure 2 caption, change DESs to DES and close the bracket at the end of the caption.
  12. Table 3 title, change to Extraction of Tocols from Deodorizer Distillate using different types of DESs.
  13. Change the title of Table 4, same as Table 3
  14. Avoid using abbreviations in Table titles and figure legends
  15. L228, revise, it is incorrect.
  16. L341, change to “could be divided”
  17. Authors should explain Type I, II, and III DESs.
  18. Authors should include a Table on comparative analysis of extraction with conventional solvents or methods.
  19. Authors should include a discussion on the economic feasibility of scaling up this method for industrial applications.
  20. A paragraph on the reusability and stability of DESs in extraction processes should be added.

Author Response

Response to Reviewer 1 Comments

1. Summary

2. Comments on the Quality of English Language

Comment : significant English language improvements are needed for better readability and clarity. Many sentences are grammatically incorrect, overly complex, or awkwardly phrased, making it difficult to follow the scientific discussion. Additionally, there are inconsistencies in terminology, verb tense, and article usage throughout the text. I have pointed out a few instances for improvement in the English language below along with other comments. However, thorough proofreading is required for the whole manuscript.

Response: Thank you for your valuable feedback regarding the clarity and readability of our manuscript. We have conducted a thorough proofreading of the entire text to improve the language quality, simplify complex sentences, and ensure consistency in terminology and verb usage. In proofreading process, author have utilized TRINKA AI, which is subscribed to by the University of Indonesia.   

3. Point-by-point response to Comments and Suggestions for Authors

Comments 1: L30, Abstract, remove in and with. It should be “in tocols extraction is presented considering three different ways”.

Response 1: Thank you for pointing this out. We agree with this comment. Therefore, We have revised it according to the suggestions from the reviewer (abstract_Page 1_Line 30)

Comments 2: L 38, Change to “phenolic structure and also briefly discusses the toxicity of DES.”

Response 2: Agree.  We have revised it (Abstract_Page 1 _ Line 38).

Comments 3: L44-45, change to “that contain a” and replace on with “in their structure”.

Response 3 :   We have revised it become “Tocols are a group of terpenoids with a chromanol ring structure and a 13-carbon side chain at the C2 position “(Page 2_Paragraph 1_Line 43-44)

Comments 4: L58, change to "These attributes make them”

Response 4 : Agree. We have revised it (Page 2_Paragraph  1_Line 58)

Comments 5: L75, change to For example

Response 5 : Agree. We have revised it (Page 2_Paragraph 3_Line 75)

Comments 6: Table 1, in the footnote, expand the full form of Mr.

Response 6 : as suggestion from reviewer 2 Mr was deleted (page 3, footnote Table 1)

Comments 7: L99-98, change to “Deodorizer distillates (DDs), derived from vegetable oil refining, are a potential source for producing tocols, considering their lower cost and higher concentration compared to vegetable oils.”

Response 7 : Agree. We have revised it (Page 3, Paragraph 1 )

Comments 8: Table 2 title, change to “Tocols content in”

Response 8 : Agree. We have revised it (Table 2, Page 3-4)

Comments 9: Be consistent with the format of et al in the manuscript, in some places, it is italicized and in some places it is normal.

Response 9 : We have revised italicized et al be normal et al. (Page 4, Paragraph 2)

Comments 10: Section 2, avoid using abbreviations in section headings

Response 10: We have revised it (Section 2 Page 5)

Comments 11: Figure 2 caption, change DESs to DES and close the bracket at the end of the caption.

Response 11: We have revised it (Page 6, Caption figure 2)

Comments 12: Table 3 title, change to Extraction of Tocols from Deodorizer Distillate using different types of DESs.

Response 12: We have revised it (Table 3, Page 6)

Comments 13: Change the title of Table 4, same as Table 3

Response 13: We have revised it (Table 4 page 7)

Comments 14: Avoid using abbreviations in Table titles and figure legends

Response 14: We have revised it (Page 8,  figure 3 legend,  Page 10, figure 4 legend, Page 15 table 5 title)

Comments 15: L228, revise, it is incorrect

Response 15 : I have revised A to α symbol (Page 6, Table 3)

Comments 16 : L341, change to “could be divided”

Response 16 : We have revised it  (Page 11 Paragraph 1)

Comments 17 : Authors should explain Type I, II, and III DESs.

Response 17 : Explanation added (Page 15, Paragraph 2)

Comments 18 : Authors should include a Table on comparative analysis of extraction with conventional solvents or methods

Response 18: The method of extracting vitamin E from deodorizer distillate using conventional methods is reported in only one article, which has been narrated in the introduction (page 4, paragraph 2), specifically using the Soxhlet method

Comments 19 : Authors should include a discussion on the economic feasibility of scaling up this method for industrial applications.

Response 19 : discussion on the economic feasibility of scaling up this method for industrial applications have been added (Page 17-18 paragraph 2)

20.         A paragraph on the reusability and stability of DESs in extraction processes should be added

Response 20: have been added (Page 18 paragraph 3)

Reviewer 2 Report

Comments and Suggestions for Authors

Review

New Horizon in Selective Tocols Extraction from Deodorizer Distillates Under Mild Conditions by using Deep Eutectic Solvents

The review presented is devoted to the study of the effectiveness toсols extraction by DES. In biotechnology, DES is used as a medium for biocatalytic reactions, for the extraction of physiologically active compounds from natural raw materials, for the pretreatment of lignocellulose biomass, and for the production of biodegradable plastics. The authors discuss the extraction process from the perspective of the intermolecular interaction between toсols and solvents, consider the possibility of using binary solvents (manifestation of a synergistic effect), process conditions (temperature conditions, mixing conditions in order to improve mass transfer characteristics), in a particular case, the issues of regeneration and aftertreatment are addressed. In this view, the material corresponds to the subject of the Journal Molecules. The paper is of practical interest because it allows summarizing some results on the application of promising DES to solve a specific technological problem.

There are few comments on the paper content, they are given below. The key question is point 1). In my opinion, it requires appropriate explanations.

Comments

1) The review and analysis of scientific publications is usually carried out by authors who have publications in the research field considered, and the authors provide references to their works (papers), which confirms the authors' qualifications in this field. The list of references in the paper presented is impressive, however, I did not find any references by the authors to their own publications (on the topic considered). What is the reason for this? Perhaps I was inattentive or there is a change of last name for some authors.

2) The authors discuss the use of DES for tocols extraction. Probably, the keywords should be supplemented with the word “extraction”.

3) Choosing a research object (natural alpha-tocopheryl acetate) on page 2, it is given in sufficient detail. Additional explanations (page 3, first paragraph « Healthy women aged 21-37 ….»), in my opinion, are unnecessary, they are not directly related to the subject. I would recommend deleting this paragraph.

4) How much detailed physical-chemical information of tocols is needed (which is shown in Table 1)?. This is publicly available information, perhaps it is worth specifying only those parameters that are needed for discussion. Also specify the conditions (pressure) under which the parameters (boiling point, melting point) are specified.

5) The vast majority of references (77 out of 119) relate to the Introduction, which explains the choice of an object (tocols) and the need to separate them using various methods and solvents. Some of the references are redundant and do not directly relate to the title of the paper.

6) Figure 2 is given earlier than its mention in the text, which is incorrect. In addition, Figure 2 is of poor quality.

7) Page 6 “Typically, this solvent is a consistent liquid formed by easily combining at least two components namely HBA and HBD that can form a single melting and freezing point lower than the melting point of its pure component” - Is it about the eutectic composition of the mixture?

8) There are many publications in the literature on the experience of using DES for flavonoid extraction. This experience can also be useful due to the similarity of properties.

Author Response

Response to Reviewer 2 Comments

1. Summary

3. Point-by-point response to Comments and Suggestions for Authors

Comments 1:  The review and analysis of scientific publications is usually carried out by authors who have publications in the research field considered, and the authors provide references to their works (papers), which confirms the authors' qualifications in this field. The list of references in the paper presented is impressive, however, I did not find any references by the authors to their own publications (on the topic considered). What is the reason for this? Perhaps I was inattentive or there is a change of last name for some authors.

Response 1: Thank you for pointing this out. We agree with this comment. Our authors have conducted extensive research in the field of extraction using DESs and ILs, as seen in the following publications :

https://doi.org/10.1016/j.ultsonch.2025.107271, https://doi.org/10.22159/ijap.2024.v16s3.09, https://doi.org/10.17957/IJAB/15.2199,  https://doi.org/10.3390/molecules29092093, https://doi.org/10.35814/jifi.v22i1.1531, https://doi.org/10.1016/j.arabjc.2023.105537, DOI: https://doi.org/10.56499/jppres23.1727_11.6.1056,  https://doi.org/10.1016/j.heliyon.2023.e20480 ,  http://dx.doi.org/10.29228/jrp.401,  https://doi.org/10.3389/fphar.2023.1012716, https://doi.org/10.56425/cma.v2i1.47, http://dx.doi.org/10.29228/jrp.401  

However, for the extraction from the deodorizer distillate matrix, we have only conducted one study, which involves the use of ionic liquids (DOI: https://dx.doi.org/10.22159/ijap.2024.v16s3.13  ). This research has already been added to the manuscript  (page 4 paragraph 2)

Comments 2: The authors discuss the use of DES for tocols extraction. Probably, the keywords should be supplemented with the word “extraction”

Response 2: Agree.  We have added it (Abstract_Page 1 _ Key words).

Comments 3: Choosing a research object (natural alpha-tocopheryl acetate) on page 2, it is given in sufficient detail. Additional explanations (page 3, first paragraph « Healthy women aged 21-37 ….»), in my opinion, are unnecessary, they are not directly related to the subject. I would recommend deleting this paragraph.

Response 3 : agree, We have deleted it  

Comments 4: How much detailed physical-chemical information of tocols is needed (which is shown in Table 1)?. This is publicly available information, perhaps it is worth specifying only those parameters that are needed for discussion. Also specify the conditions (pressure) under which the parameters (boiling point, melting point) are specified.

Response 4 : We have deleted Mr and appearance data and give fotenote pressure for boiling point and melting point (Page 3_table 1)

Comments 5: The vast majority of references (77 out of 119) relate to the Introduction, which explains the choice of an object (tocols) and the need to separate them using various methods and solvents. Some of the references are redundant and do not directly relate to the title of the paper.

Response 5 :

Some of the references that are redundant and do not directly relate to the title of the paper have been deleted.

We have deleted ref 40 from page 4 paragraph 1

We have deleted ref 60 from page 4 paragraph 3

We have deleted ref 61 from page 4 paragraph  3

We have deleted ref 64 from page 5 paragraph  1

We have deleted ref 68-69 from page 5 paragraph  1

We have deleted ref 71-749 from page 5 paragraph  1

Comments 6: ) Figure 2 is given earlier than its mention in the text, which is incorrect. In addition, Figure 2 is of poor quality.

Response 6 : Figure 2 has given after than its mention in the text (page 6_Figure 2)

"The number in the lower left corner of the image has been cut off and has been corrected, and the clarity of the image has been improved. As for the right image (NOESY), it is original from the journal that has been cited."

Comments 7: Page 6 “Typically, this solvent is a consistent liquid formed by easily combining at least two components namely HBA and HBD that can form a single melting and freezing point lower than the melting point of its pure component” - Is it about the eutectic composition of the mixture?

Response 7 : yes right, This sentence is to describe what deep eutectic solvents are

Comments 8: There are many publications in the literature on the experience of using DES for flavonoid extraction. This experience can also be useful due to the similarity of properties

Response 8 : Thank you very much for the excellent suggestions. In this review, we have also made an effort to discuss compounds that belong to the same group as vitamin E, that is polyphenols, although it is not specifically focused on the flavonoid as one of class of polyphenols